# *GSTM5* as a Potential Biomarker for Treatment Resistance in Prostate Cancer

**DOI:** 10.3390/biomedicines13081872

**Published:** 2025-08-01

**Authors:** Patricia Porras-Quesada, Lucía Chica-Redecillas, Beatriz Álvarez-González, Francisco Gutiérrez-Tejero, Miguel Arrabal-Martín, Rosa Rios-Pelegrina, Luis Javier Martínez-González, María Jesús Álvarez-Cubero, Fernando Vázquez-Alonso

**Affiliations:** 1Biochemistry and Molecular Biology III and Immunology Department, Faculty of Medicine, University of Granada, PTS, 18016 Granada, Spain; pmporras@ugr.es (P.P.-Q.); luciachica@ugr.es (L.C.-R.); luisjavier.martinez@genyo.es (L.J.M.-G.); mjesusac@ugr.es (M.J.Á.-C.); 2GENYO, Centre for Genomics and Oncological Research-Pfizer, University of Granada, Andalusian Regional Government, PTS, 18016 Granada, Spain; 3Legal Medicine and Toxicology Department, Faculty of Medicine, University of Granada, PTS, 18016 Granada, Spain; 4Department of Urology, San Cecilio University Hospital, 18007 Granada, Spain; franciscogutej@gmail.com (F.G.-T.); arrabalm@ugr.es (M.A.-M.); 5Biosanitary Research Institute (ibs. GRANADA), University of Granada, PTS, 18012 Granada, Spain; rosa.rios.pelegrina@gmail.com; 6Department of Pathology, University Hospital Virgen de las Nieves, 18014 Granada, Spain; 7Urology Department, University Hospital Virgen de las Nieves, 18014 Granada, Spain; fvazquezalonso@gmail.com

**Keywords:** castration-resistant prostate cancer, *GSTM5*, biomarker

## Abstract

**Background/Objectives**: Androgen deprivation therapy (ADT) is widely used to manage prostate cancer (PC), but the emergence of treatment resistance remains a major clinical challenge. Although the GST family has been implicated in drug resistance, the specific role of *GSTM5* remains poorly understood. This study investigates whether *GSTM5*, alone or in combination with clinical variables, can improve patient stratification based on the risk of early treatment resistance. **Methods**: In silico analyses were performed to examine *GSTM5*’s role in protein interactions, molecular pathways, and gene expression. The rs3768490 polymorphism was genotyped in 354 patients with PC, classified by ADT response. Descriptive analysis and logistic regression models were applied to evaluate associations between genotype, clinical variables, and ADT response. *GSTM5* expression related to the rs3768490 genotype and ADT response was also analyzed in 129 prostate tissue samples. **Results**: The T/T genotype of rs3768490 was significantly associated with a lower likelihood of early ADT resistance in both individual (*p* = 0.0359, Odd Ratios (OR) = 0.18) and recessive models (*p* = 0.0491, OR = 0.21). High-risk classification according to D’Amico was strongly associated with early progression (*p* < 0.0004; OR > 5.4). Combining genotype and clinical risk improved predictive performance, highlighting their complementary value in stratifying patients by treatment response. Additionally, *GSTM5* expression was slightly higher in T/T carriers, suggesting a potential protective role against ADT resistance. **Conclusions**: The T/T genotype of rs3768490 may protect against ADT resistance by modulating *GSTM5* expression in PC. These preliminary findings highlight the potential of integrating genetic biomarkers into clinical models for personalized treatment strategies, although further studies are needed to validate these observations.

## 1. Introduction

Prostate cancer (PC) treatment resistance represents a significant clinical challenge, affecting not only therapeutic efficacy but also patient survival. Androgen deprivation therapy (ADT) is one of the mainstays of treatment for PC, particularly for cases where the tumor is not localized [1]. ADT effectively reduces testosterone levels, an androgenic hormone essential for the growth of prostate cells. By lowering testosterone, AR (androgen receptor) stimulation is diminished, which in turn reduces PSA (prostate-specific antigen) levels and halts tumor growth [2]. However, despite its initial effectiveness, more than half of the patients eventually progress to castration-resistant prostate cancer (CRPC), in which tumor growth persists and PSA levels increase despite reduced testosterone levels [1,3,4]. It is important to note that there is still some controversy regarding the exact proportion of patients who develop resistance, as this largely depends on the timing and criteria used to assess it. Nevertheless, some studies suggest that, over time, nearly all patients undergoing long-term ADT ultimately develop resistance [5]. This resistance is associated with limited long-term survival, as the median overall survival for patients with metastatic CRPC is generally less than two to three years [4,6]. The emergence of CRPC underscores the need to characterize patient subgroups who may benefit from alternative or complementary therapeutic approaches [7]. In this context, and considering the clinical impact of treatment resistance in PC, risk stratification plays a crucial role in therapeutic decision-making [8,9]. Currently, the D’Amico risk classification is widely used to categorize patients with PC based on baseline PSA levels, Gleason Score, and clinical stage. However, its predictive power could be significantly enhanced by integrating genetic biomarkers and advanced imaging [10]. In particular, identifying genetic variants associated with ADT resistance could provide novel opportunities for more personalized and effective therapies [11]. For instance, alterations in genes such as *RB1*, *TP53*, and *PTEN*, as well as increased *MYCN* amplification, have been identified as key drivers in the development of neuroendocrine PC (NEPC), a molecular PC subtype that emerges after ADT failure [12].

In line with these advancements, several studies have demonstrated that the AR can be activated independently of androgens, allowing tumor progression to continue despite hormonal deprivation. This phenomenon suggests that resistance to ADT is not solely dependent on androgen levels but is instead mediated by a range of alternative mechanisms that enable the AR to maintain its function in driving tumor progression [2,13,14], including those driven by oxidative stress [15,16]. In PC, oxidative stress has been linked to tumor initiation and progression, as well as to the development of CRPC, in part through activation of AR signaling pathways. However, the precise molecular interplay between oxidative stress and AR signaling remains poorly understood [15].

It has been shown that redox balance in advanced tumor cells may be mediated by the glutathione (GSH) system, enabling tumor cells to evade oxidative stress-induced cell death [17]. In this context, members of the glutathione S-transferase (GST) family, including *GSTM5*, are thought to play a key role in the detoxification of free radicals and the regulation of redox status in PC [18]. Interestingly, *GSTM2* has been shown to contribute to resistance against second-generation androgen receptor inhibitors (SG-ARIs), such as enzalutamide, apalutamide, and darolutamide [19]. Specifically, it has been shown that *GSTM2* overexpression in enzalutamide-resistant cells contributes to oxidative stress damage evasion and activates the p38 MAPK pathway. Furthermore, glutathione reductase activity has been described as a key factor in oxidative stress adaptation in metastatic PC cells, providing resistance to peroxide-induced cytotoxicity and facilitating tumor survival under adverse therapeutic conditions [17]. Similarly, Zhang et al. observed a significant correlation between AR expression and *GSTM1* copy number, suggesting a functional interaction between AR regulation and GSH-dependent detoxification pathways in PC [20].

Regarding *GSTM5,* its expression has been shown to be significantly downregulated in various cancer types, including bladder, ovarian, and lung cancer [21,22]. For instance, in bladder cancer, *GSTM5* overexpression has been linked to reduced cell proliferation, migration, and colony formation [21], while in lung adenocarcinoma, aberrant *GSTM5* expression correlates with poor overall survival [22]. These findings suggest that *GSTM5* may also represent a potential therapeutic target in certain cancer contexts, particularly through redox-modulating or epigenetic-based strategies such as demethylating agents, although its role in PC remains largely unexplored.

Despite the established relevance of *GSTM5* in cancer, its specific function in PC and treatment response remains poorly understood. This study, therefore, focuses on exploring the impact of *GSTM5* on ADT resistance in patients with PC. By examining both *GSTM5* expression levels and genetic variants, we aim to gain valuable insights into its role in modulating tumor response to ADT, potentially offering new therapeutic strategies for treatment resistance and improving the clinical management of advanced PC [19]. This effort not only has the potential to improve risk stratification but also to guide the use of novel targeted therapies, which are essential for managing this type of cancer more effectively and improving patient quality of life and survival.

## 2. Methods

### 2.1. In Silico Analysis of GSTM5 Role in PC

In order to investigate the functional role of *GSTM5* in PC, a protein–protein interaction (PPI) network was constructed using STRING (v12.0) with a high-confidence interaction threshold (0.7), selecting the top 15 interactors. To identify significantly enriched pathways, functional enrichment analysis was performed using GeneCodis (v4.0) [23], integrating KEGG, Gene Ontology, and InterPro. The results were filtered using a false discovery rate (FDR) threshold of 0.05. Additionally, DisGeNET (v24.1) was used to assess the association of *GSTM5* and its top 15 interactors with prostate neoplasm and malignant prostate neoplasm. To further explore the role of *GSTM5* in PC, its expression levels were analyzed using The Cancer Genome Atlas (TCGA, n = 549), comparing tumor vs. normal prostate tissue and across different tumor aggressiveness stages. Furthermore, promoter methylation of *GSTM5* was investigated in TCGA data to examine potential epigenetic regulation. The methylation levels in the *GSTM5* promoter region were compared between tumor and normal prostate tissue, and the results were analyzed to identify potential correlations between *GSTM5* silencing through hypermethylation and advanced PC progression. The *GTSM5* expression analysis was expanded using the Prostate Cancer Atlas (PCA, n = 1365), incorporating multiple independent cohorts (GSE21034, MSKCC, SU2C/PCF, and E-MTAB-6128) for validation.

### 2.2. In Silico Characterization of the rs3768490 Variant (GSTM5)

The rs3768490 (G > T) polymorphism in *GSTM5* was analyzed in silico to assess its potential functional impact. MutationTaster (v2.0, [24]) was used to predict effects on transcriptional regulation, mRNA processing, and potential splicing site alterations. Additionally, population databases (1000 Genomes, ExAC) were consulted to determine SNP frequency and evolutionary conservation scores (phyloP, phastCons). Splicing site alterations were also evaluated using MutationTaster, which provided scores indicative of potential changes in the splicing donor and acceptor sites.

### 2.3. Study Design

The present study included a cohort of 354 Caucasian patients diagnosed with PC. The diagnosis was established based on serum PSA levels > 4 ng/dL and histologically confirmed prostatic adenocarcinoma. A biological sample was obtained from each patient, consisting of peripheral blood and/or buccal swab, which were used for genetic analysis. Blood samples were processed by centrifugation at 1400 rpm for 10 min at 4 °C, separating the cellular fraction and plasma, which were stored independently at −80 °C until further use. Buccal swabs were stored at −20 °C. Additionally, prostate tissue samples were obtained from 129 patients with PC, consisting of fresh prostate tissue, which was immediately frozen and stored at −80 °C. These samples were later used for gene expression studies.

For all patients, key clinical variables were collected, including age, diagnostic PSA, Gleason score, ISUP grade, D’Amico risk classification, TNM staging, and treatment received. Additionally, the presence of metastases and mortality were recorded when applicable. Further details are shown in Table 1.

Within this cohort, a total of 222 patients underwent ADT and met the criteria for the analysis of treatment response. Patients were classified into three groups based on their clinical evolution: *Early-onset CRPC* (patients who develop resistance to ADT within the first three years of treatment), *Long-Term responders* (patients who have been on ADT for at least three years and have not developed resistance up to the current date), and *Non-progression responders* (a subgroup of *Long-term responders* who remained responsive to ADT beyond five years of treatment). The criterion used to define treatment resistance was three consecutive increases in PSA serum levels or the development of metastases during ADT.

All study participants provided written informed consent before being enrolled, and the study was previously approved by the Research Ethics Committee of Granada Center (CEI-Granada internal code 1638-N-18) in accordance with the Declaration of Helsinki.

### 2.4. DNA Extraction from Blood Samples and Buccal Swabs

Genomic DNA was extracted from peripheral blood samples using the Real Blood DNA kit (Real Laboratory, Valencia, ES), following the manufacturer’s instructions. For buccal swabs, DNA extraction was performed as described by Freeman et al. [25] with additional optimizations developed by Gómez-Martín et al. [26]. The quality and quantity of extracted DNA were assessed using a NanoDrop 2000c (ThermoFisher Scientific, Waltham, MA, USA) to ensure its suitability for downstream applications. The extracted DNA was stored at −20 °C until genotyping studies.

### 2.5. Genotyping Analysis by qPCR

DNA genotyping of rs3768490 was performed using a specific TaqMan probe (C___3184500_20; ThermoFisher Scientific, Waltham, MA, USA) in combination with TaqMan Genotyping Master Mix (ThermoFisher Scientific, Waltham, MA, USA). Allelic discrimination assays were conducted in 384-well plates using the QuantStudio 6 Flex Real-Time PCR System (Applied Biosystems, Waltham, MA, USA). The qPCR reaction was performed as follows: 60 °C for 30 s and 95 °C for 10 min; followed by 40 cycles of 15 s at 95 °C and 1 min at 60 °C. A negative template control (NTC) was included in each plate.

### 2.6. RNA Extraction and Reverse Transcription from Fresh Prostatic Tissue

Total mRNA was extracted from fresh prostate tissue samples using the Trizol^®^/chloroform method following standard protocols. RNA quality and purity were assessed by measuring the A260/A280 ratio using a NanoDrop 2000c spectrophotometer (ThermoFisher Scientific, Waltham, MA, USA). The extracted RNA was stored at −80 °C to ensure stability until further processing. All samples were normalized to 100 ng/µL. Reverse transcription was performed using the PrimeScript RT Reagent Kit (Takara Bio, Kusatsu, Japan), following the manufacturer’s instructions. The cDNA obtained was stored at −20 °C until gene expression analysis.

### 2.7. Gene Expression Analysis by qPCR

Gene expression analysis was conducted using SYBR Green designed probes (Life Technologies, Carlsbad, CA, USA), on a 96-well plate with the QuantStudio 6 Flex Real-Time PCR System (Applied Biosystems, Waltham, MA, USA). The qPCR reaction was performed as follows: 95 °C for 10 min for enzyme activation; followed by 40 cycles of 15 s at 95 °C and 1 min at 60 °C for denaturation and annealing/extension. Primers for *GSTM5* and *HPRT1,* the latter used as housekeeping control, were designed using Primer-Blast (NIH) software v2.5.0 [27] and supplied by Sigma Aldrich (San Luis, MO, USA) with the following sequences: *GSTM5* (forward): CTCTCAAAGTCTGAGCCCCG; *GSTM5* (reverse): GTCATAGTCAGGAGCGTCCC; *HPRT1* (forward): TTATGGACAGGACTGAACGTCTT; *HPRT1* (reverse): CTTGAGCACACAGAGGGCTA.

All samples were analyzed in triplicate, and each plate included an NTC. A common threshold was set for each gene, and threshold cycle (Ct) values were obtained for each sample. A Ct ≥ 35 was considered undetermined. The expression level of *GSTM5* was quantified using the relative quantification (RQ) method, normalized to the housekeeping gene *HPRT1*.

### 2.8. Statistical Analysis

A comprehensive statistical analysis was conducted, integrating both descriptive and inferential approaches. All statistical procedures were executed in R (v4.4.1) using the stats (v4.4.1) and MASS (v7.3-61).

The association of treatment response in patients with PC with the rs3768490 genotype and disease classification was analyzed by chi-square tests (χ^2^) or Fisher’s exact test for small sample sizes. To strengthen the analysis, simple logistic regression was performed to quantify the strength and direction of these associations beyond mere statistical significance. Results were expressed as Odds Ratios (OR) with 95% confidence intervals (95% CI), and statistical significance was set at *p* < 0.05. This combined approach facilitated the selection of relevant variants for subsequent analyses.

To enhance the predictive accuracy of the models, the D’Amico risk classification—an established prognostic tool that integrates PSA levels, clinical stage, and Gleason score into a single risk measure—was incorporated alongside genotyping data for multiple logistic regression analysis. Stepwise selection based on the Akaike Information Criterion (AIC) was employed to confirm that both the genotypic and clinical classifications contributed meaningfully to patient stratification. Models were accepted at *p* < 0.05.

For the *GSTM5* expression analysis, data normality was assessed using the Shapiro–Wilk test. The median and 95% confidence intervals were calculated for each group, and the Kruskal–Wallis test and Mann–Whitney U test were used to identify significant differences. These analyses were performed using GraphPad Prism (v8.0.1).

## 3. Results

### 3.1. Expression and Functional Role of GSTM5 in PC

To explore the functional role of *GSTM5* in PC, a PPI network was constructed using STRING with a high-confidence threshold of 0.7 to identify the top 15 interactors. Among the identified proteins, three major pathways were highlighted, involving carcinogenesis, drug metabolism via cytochrome P450, and other enzymatic detoxification systems, all of which included *GSTM5*. A visual representation of the PPI network, highlighting the functional classification of genes, is shown in Figure 1A. Furthermore, functional enrichment analysis of these interacting genes revealed significant KEGG pathways closely related to cancer progression and treatment response, including chemical carcinogenesis, drug metabolism, and pathways in cancer, among others (Figure 1B, Appendix A).

To further evaluate the clinical relevance of these interactors, disease association analysis was performed using DisGeNET. The results showed that the identified genes were significantly associated with prostatic neoplasm (FDR = 2.1 × 10^−2^) and malignant prostate neoplasm (FDR = 2.1 × 10^−2^). These findings suggest that *GSTM5* and its interacting partners may play a crucial role in tumorigenesis and drug metabolism in PC, potentially affecting treatment response and resistance mechanisms.

To investigate the expression levels of *GSTM5* in PC, data from TCGA were analyzed. The results revealed that *GSTM5* was significantly downregulated in tumor tissue compared to normal prostate tissue (*p* < 0.0001), suggesting a potential tumor suppressor role (Figure 1C). Furthermore, when patients were stratified based on tumor aggressiveness, *GSTM5* expression remained consistently downregulated, with a progressive decrease as tumor aggressiveness increased (Figure 1D). These findings indicate that *GSTM5* downregulation may be associated with PC progression, reinforcing its potential involvement in tumor suppression and disease severity. TCGA data were also used to analyze methylation changes in the *GSTM5* promoter sequence in PC. Our results revealed that the *GSTM5* promoter is significantly hypermethylated in tumor samples compared to normal prostate tissue (*p* = 1.9 × 10^−12^, Figure 1E). This suggests that epigenetic regulation through hypermethylation may contribute to *GSTM5* silencing in PC, potentially promoting tumor progression and treatment resistance.

To further validate these findings, we repeated the analysis using data from the PCA, which integrates TCGA with additional cohorts, including GSE21034, MSKCC, SU2C/PCF, and E-MTAB-6128. This expanded dataset allowed us to examine a broader spectrum of molecular subtypes in PC. The results confirmed that all molecular subtypes of PC showed lower *GSTM5* expression compared to normal prostate tissue, supporting the notion that *GSTM5* downregulation is a common feature across different forms of PC (Figure 1F).

### 3.2. Functional Impact of rs3768490 (GSTM5)

The in silico analysis using MutationTaster indicated that rs3768490 is located in the 3′UTR of *GSTM5*. Although it does not alter the protein sequence, it may affect post-transcriptional regulation. The SNP was associated with H3K27me3 epigenetic modification, suggesting a possible role in gene repression. Moreover, the variant increased the strength of an existing splicing donor site (wild-type score: 0.45; mutant score: 0.76) and led to the gain of a novel donor site (score: 0.34), which could influence mRNA stability and alternative splicing. Its low evolutionary conservation (phyloP = −0.339, phastCons = 0) suggests that it is not highly preserved across species, although it may have a regulatory role in PC.

### 3.3. Association of rs3768490 Genotype with ADT Response

To evaluate the association between clinical variables and ADT response in patients with PC, two grouping strategies were applied: (i) *Early-onset CRPC* (n = 54) vs. *Long-term responders* (n = 67), and (ii) *Early-onset CRPC* vs. *Non-progression responders* (n = 64). The same grouping was used to analyze genotype distribution according to treatment response.

Descriptive analysis revealed a strong association between clinical variables, except PSA, and treatment response in *Long-term responders* (Appendix A) and *Non-progression responders* (Appendix A) compared to *Early-onset CRPC* patients. In particular, patients with Gleason scores ≥ 7, tumor stage T3–T4, and high-risk classification according to D’Amico showed a greater likelihood of developing early resistance to ADT (Figure 2A,B). These findings indicate that patients with more adverse clinical features at diagnosis have a significantly higher probability of developing treatment resistance within the first three years of treatment.

Regarding genotype analysis, the T/T genotype showed a significant association with a lower probability of belonging to the *Early-onset CRPC* group compared to the *Long-term responders* group (*p* = 0.0359 in simple logistic regression; OR = 0.18, 95% CI: 0.02–0.76). The recessive model (TT vs. GG + GT) also showed a trend toward significance (*p* = 0.0636 in Fisher’s test), with a reduced risk of early progression in T/T genotype carriers (*p* = 0.0572 in simple logistic regression; OR = 0.22, 95% CI: 0.03–0.88) compared to those with GG + GT genotypes. Multiple logistic regression analysis confirmed this association, with an OR of 0.18 (95% CI: 0.02–0.86; *p* = 0.0487) for the T/T genotype and an OR of 0.21 (95% CI: 0.03–0.96; *p* = 0.0604) in the recessive model (Figure 2C and Appendix A).

A similar trend was observed when comparing the *Early-onset CRPC* and *Non-progression responders* groups (Figure 2B). Patient with the T/T genotype showed a significantly lower probability of developing treatment resistance within five years (*p* = 0.0359; OR = 0.18, 95% CI: 0.02–0.76). The recessive model also showed a significant association (*p* = 0.0491; OR = 0.21, 95% CI: 0.03–0.83), supporting the lower risk of early resistance for the T/T carriers. Multiple logistic regression confirmed these findings, with an OR of 0.18 (95% CI: 0.02–0.86, *p* = 0.0484) for the T/T genotype and an OR of 0.19 (95% CI: 0.03–0.86; *p* = 0.0509) in the recessive model (Figure 2C and Appendix A).

In both analysis models, high-risk classification according to D’Amico was a significant predictor of early progression to CRPC, with OR values greater than 5 in all cases (*p* < 0.001) (Figure 2C and Appendix A).

These results suggest that the T/T genotype of the rs3768490 SNP in *GSTM5* may be associated not only with a lower risk of early resistance to ADT, but also with sustained sensitivity to ADT, potentially preventing the development of CRPC in patients with PC. Further studies are needed to confirm its role as a prognostic biomarker.

### 3.4. Impact of rs3768490 on GSTM5 Expression and ADT Outcome

*GSTM5* expression levels were analyzed in relation to rs3768490 genotype, D’Amico risk classification, and response to ADT (Appendix A). *GSTM5* expression was higher in T/T carriers, both when considering the three genotypes separately and under the recessive model (Figure 3A,B). However, this difference did not reach statistical significance, possibly due to the low frequency of the T/T genotype. Additionally, *GSTM5* expression was lower in high-risk patients according to the D’Amico classification (Figure 3C). Although this decrease was not statistically significant, it is consistent with the genotyping results, where the T/T genotype was associated with a better prognosis, potentially due to higher *GSTM5* expression levels.

Regarding ADT response, *Early-onset CRPC* patients exhibited lower *GSTM5* expression levels than *Long-term responders* (Figure 3D). This difference was even more pronounced when comparing *Early-onset CRPC* patients with *Non-progression responders* (Figure 3E). These findings suggest a potential trend toward increased *GSTM5* expression in patients with a prolonged response to ADT.

ROC curve analysis was performed to evaluate the predictive ability of *GSTM5* expression in distinguishing *Early-onset CRPC* from ADT responders (Figure 3F). The model showed a moderate discriminative ability when *Early-onset CRPC* was compared with *Long-term responders* (AUC = 0.7813), and improved discriminative ability when compared with *Non-progression responders* (AUC = 0.8214).

## 4. Discussion

GSH metabolism plays a fundamental role in the cellular response to therapy, particularly in xenobiotic detoxification, oxidative stress protection, and drug resistance promotion [28,29]. In this context, *GSTM5* may contribute to modulating treatment response in PC through several mechanisms. First, *GSTM5* regulates cellular redox balance by influencing intracellular GSH availability, a critical metabolite that protects cells against oxidative stress and chemotherapy-induced damage [21]. Second, its involvement in resistance to platinum-based drugs suggests that *GSTM5* expression levels could affect the efficacy of these treatments in advanced PC [30]. Additionally, *GSTM5*’s interaction with DNA repair mechanisms may influence activation of the NRF2 pathway, thereby impacting the tumor cell’s ability to respond to treatment-induced damage [31].

Despite the well-established role of *GSTM5* in oxidative stress modulation and drug resistance mechanisms, the impact of its genetic variants in the context of PC remains poorly understood. Other genetic variants in GST family members have been associated with increased PC risk. For example, the rs1138272 polymorphism in *GSTP1* is linked to a nearly five-fold increase in risk among carriers of the variant allele [32], and the *GSTM1* null genotype has been shown to significantly increase PC risk in Asian populations [33]. Interestingly, genetic variants within the GSTM gene cluster—particularly in *GSTM3*—have been associated with cancer susceptibility in breast cancer, where certain *GSTM3* SNPs conferred increased risk only in individuals lacking *GSTM1* expression [34]. However, to date, no studies have evaluated the role of *GSTM5* genetic variants in PC. Our study is the first to assess the influence of rs3768490 polymorphism in the *GSTM5* on ADT response. The results suggest that the T/T genotype may play a protective effect against ADT resistance, as carriers of this genotype exhibited a significantly lower likelihood of early treatment failure (*p* < 0.05; OR < 0.2). Conversely, D’Amico’s high-risk classification was consistently associated with early progression (*p* < 0.0004; OR > 5.4), reinforcing its clinical value as a predictor of poor outcomes. The combination of genotype and clinical risk improved predictive capacity, emphasizing the potential utility of integrating genetic and clinical factors in ADT response stratification.

Beyond genetic associations, *GSTM5* expression levels were also evaluated. Previous studies have shown that *GSTM5* expression can be epigenetically regulated through promoter hypermethylation, leading to gene silencing and increased tumor aggressiveness. In breast cancer, *GSTM5* downregulation via promoter methylation has been linked to increased cell proliferation and migration [35]. Similarly, in bladder cancer, *GSTM5* acts as a tumor suppressor, and its promoter hypermethylation results in decreased expression, reduced glutathione levels, and enhanced tumor progression [21]. In lung adenocarcinoma, *GSTM5* promoter hypermethylation has also been identified as a key gene-silencing mechanism, correlating with poor prognosis and reduced survival. Notably, treatment with demethylating agents such as 5-Aza-CdR has been shown to restore GSTM5 expression and inhibit tumor cell proliferation and migration, reinforcing the potential therapeutic value of targeting DNA methylation in cancer. This effect is not only epigenetic but also functional, as restoration of GSTM5 expression has been associated with reduced proliferation, migration, and improved oxidative balance through GSH metabolism [25]. Demethylating agents like 5-Aza-CdR could offer a potential strategy for reactivating GSTM5 expression in PC, improving the therapeutic response and overcoming resistance. While these findings suggest that *GSTM5* expression may be epigenetically regulated by methylation, our results indicate that it may also be influenced by specific genetic variants, such as rs3768490. This is consistent with the fact that rs3768490 lies within a distal enhancer-like signature region (EH38E1375040), a regulatory element previously validated as capable of modulating gene expression [36]. These preliminary findings provide new insight into how regulatory genetic variants may modulate *GSTM5* expression and, consequently, influence treatment response in PC.

From a clinical perspective, incorporating the *GSTM5* rs3768490 genotype into predictive resistance models of ADT resistance could improve patient stratification and support a more personalized therapeutic approach. Specifically, individuals carrying the G allele (G/G or G/T genotypes) may represent a subgroup with increased vulnerability to early progression and could benefit from closer monitoring or the consideration of additional therapeutic strategies. Although platinum-based agents and redox-targeted therapies are not standard in PC management, their potential role in genetically defined subgroups with altered *GSTM5* function, as well as the combination with demethylating agents, merits further investigation in preclinical and translational studies [21,30]. Of particular interest, redox modulation strategies may offer therapeutic opportunities in tumors with altered *GSTM5* function. For example, depleting intracellular glutathione using buthionine sulfoximine (BSO) has been shown to enhance the efficacy of histone deacetylase inhibitors in resistant breast cancer cells, suggesting that glutathione buffering contributes to drug resistance [37]. In ovarian cancer, *GSTM5* expression has been associated with stemness features and therapeutic response. Notably, treatment with investigational anticancer agents such as AICAR, an AMPK activator, and PI-103, a dual PI3K/mTOR inhibitor, was shown to restore *GSTM5* expression and increase treatment sensitivity in resistant ovarian cancer cells [38]. Although direct evidence in PC is still limited, the genetic and molecular similarities between prostate, breast, and ovarian cancers support further exploration of redox and epigenetic therapies in PC, particularly in molecularly defined subgroups with *GSTM5* dysregulation [39,40].

Despite the promising findings of this study, certain limitations must be considered. The small sample size limits the statistical robustness of some analyses; therefore, future studies with larger cohorts and external validation are needed to confirm the clinical relevance of *GSTM5* as a predictive biomarker of ADT resistance. Additionally, although our data suggests its involvement in treatment resistance, functional studies in cellular and preclinical models will be required to elucidate the underlying molecular mechanisms [21]. Specifically, future in vivo studies should focus on investigating how *GSTM5* interacts with key resistance pathways, such as oxidative stress regulation and AR signaling [2,13,14,15,16]. Moreover, treatment resistance in PC is a multifactorial process involving diverse molecular pathways; thus, it will be crucial to investigate how *GSTM5* interacts with other resistance pathways, such as alterations in *BRCA2*, *TP53*, and *PTEN*, which have been shown to influence therapeutic response [30]. Additionally, further functional studies are necessary to determine how GSTM5 modulates oxidative stress and its crosstalk with other signaling pathways, such as *NRF2* or DNA repair mechanisms, in the context of PC and ADT resistance [31]. Although our study focuses on *GSTM5*, it is part of the larger GST family, which includes *GSTM1* and *GSTM2*, both of which have been implicated in tumor progression and treatment response [19,20]. This underscores the importance of considering other members of the GST family in future studies that adopt more integrative approaches. In clinical practice, genotyping this variant before initiating ADT could help physicians identify patients more likely to benefit from standard hormonal therapy versus those who may require early intensification strategies, including the addition of second-line agents. These preliminary findings lay the groundwork for future clinical trials that incorporate *GSTM5* status as a stratification variable or therapeutic target.

From a translational perspective, these preliminary findings open new avenues for personalizing the management of advanced PC. In clinical practice, genotyping the rs3768490 polymorphism prior to initiating ADT could assist physicians in identifying patients more likely to benefit from standard hormonal treatment versus those who may require early intensification strategies, such as the addition of second-line agents. Prospective studies are warranted to evaluate whether *GSTM5* genotyping can be effectively integrated into clinical workflows, enabling earlier identification of patients at higher risk of developing resistance to ADT.

## 5. Conclusions

This study highlights the potential role of the *GSTM5* rs3768490 T/T genotype as a protective factor against resistance to ADT in PC. Integrating this genetic marker with clinical risk factors, such as the D’Amico classification, significantly improved the ability to identify patients at higher risk of treatment failure. In parallel, the reduced expression of *GSTM5* in tumor tissue, along with its established involvement in oxidative stress regulation and drug metabolism, support its biological relevance in therapeutic response. Together, these findings suggest that *GSTM5* may serve as a promising biomarker for refining risk stratification and guiding personalized treatment strategies. Specifically, integrating the rs3768490 genotype with established clinical classifiers could support more tailored decision-making for patients with PC undergoing ADT. In addition, the biological role of *GSTM5* in oxidative stress regulation and its potential for pharmacological modulation open the door to future therapeutic avenues involving redox-targeted therapies. Further functional and clinical validation is warranted to confirm its utility and explore potential therapeutic interventions targeting redox balance and epigenetic regulation in PC.

## Figures and Tables

**Figure 1 biomedicines-13-01872-f001:**
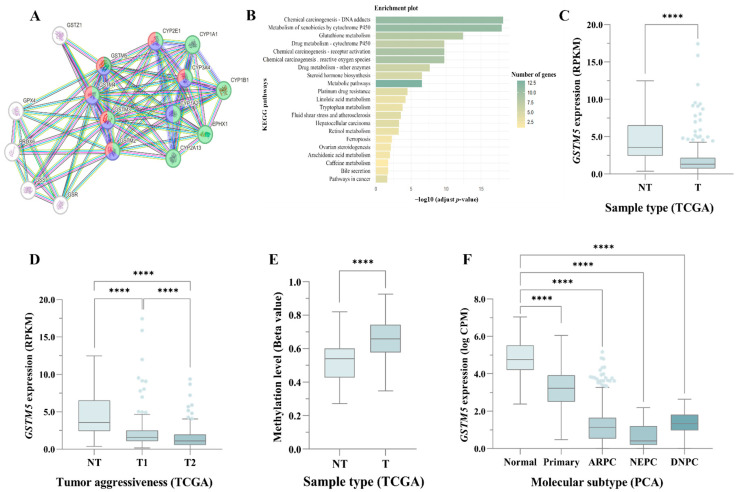
**Functional and Expression Analysis of *GSTM5* in PC.** (**A**) PPI network of *GSTM5* and its top 15 interactors, highlighting key genes involved in chemical carcinogenesis (green) and drug metabolism via cytochrome P450 (blue) or other enzymes (red). (**B**) Functional enrichment analysis of the top 15 *GSTM5* interactors. (**C**,**D**) TCGA expression data analysis across different sample types (**C**) and tumor aggressiveness stages (**D**). (**E**) TCGA promoter methylation analysis of *GSTM5* in tumoral versus normal prostate tissue. (**F**) PCA expression data analysis across different PC molecular subtypes. NT = Non-tumoral prostate sample, T = tumoral prostate sample, T1 = low-aggressive tumoral prostate sample, T2 = high-aggressive tumoral prostate sample, ARPC = androgen receptor-positive PC, NEPC = neuroendocrine PC, DNPC = double-negative PC. **** *p* ≤ 0.0001.

**Figure 2 biomedicines-13-01872-f002:**
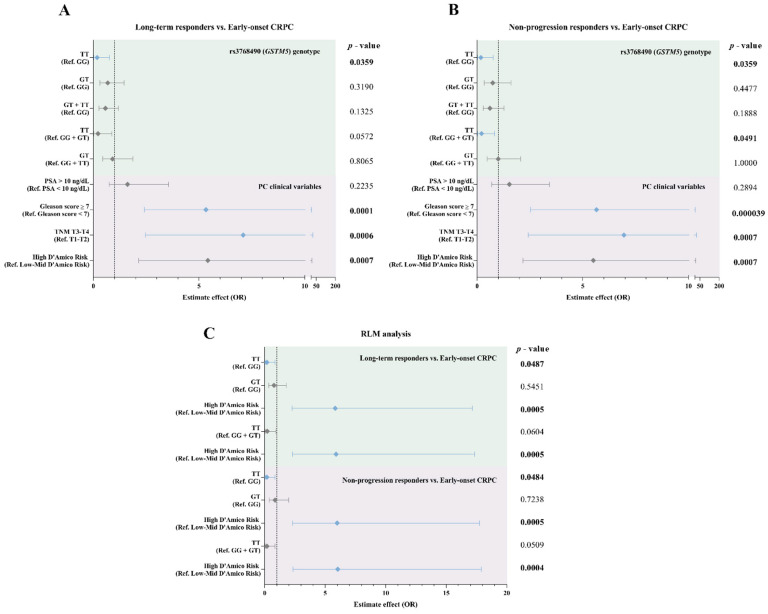
Forest plots of associations between clinical variables, ADT response, and rs3768490 genotype (*GSTM5)* in patients with PC. (**A**) Comparison of *Long-term responders* vs. *Early-onset CRPC* based on the rs3768490 genotype. (**B**) Comparison of *Non-progression responders* vs. *Early-onset CRPC* based on the rs3768490 genotype. (**C**) RLM analysis of clinical variables, comparing *Long-term responders* vs. *Early-onset CRPC* and *Non-progression responders* vs. *Early-onset CRPC*. *p*-values for each comparison are shown, with confidence intervals and effect estimates. OR = Odds Ratio. Gray bars represent non-significant associations; blue bars indicate statistically significant associations (*p* ≤ 0.05).

**Figure 3 biomedicines-13-01872-f003:**
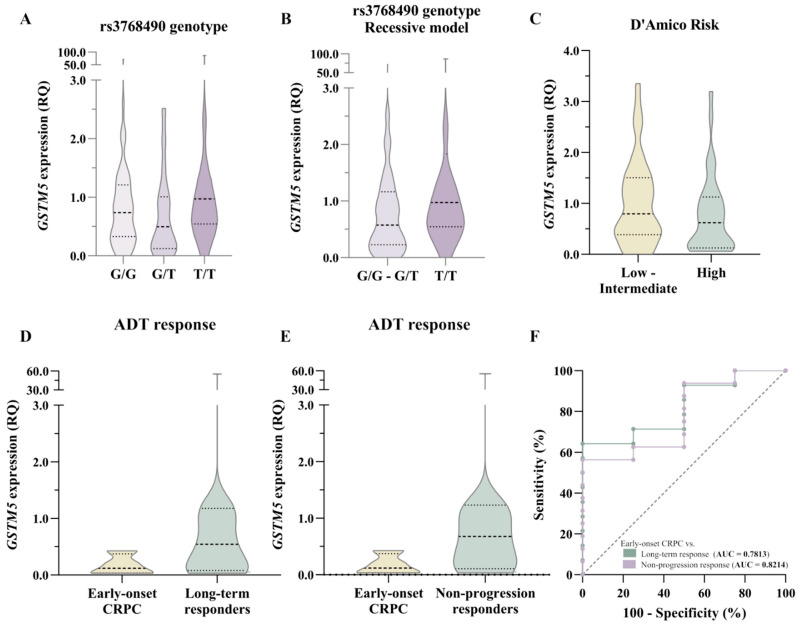
**Statistical analysis of *GSTM5* expression.** (**A**,**B**) *GSTM5* expression analysis across three rs3768490 genotypes (**A**) and in the recessive model (**B**). (**C**) *GSTM5* expression concerning D’Amico risk classification. (**D**,**E**) *GSTM5* expression in patients with *Early-onset CRPC* vs. *Long-term* (**D**) and *Non-progression responders*. (**F**) ROC curve comparing *Early-onset CRPC* vs. *Long-term* and *Non-progression responders.* RQ = Relative quantification, AUC = Area under the curve.

**Table 1 biomedicines-13-01872-t001:** Descriptive summary of clinical variables in the PC cohort.

Clinical Variable	N (%)
PSA serum levels:	
PSA < 10 ng/dL	135 (40.79%)
PSA ≥ 10 ng/dL	196 (59.21%)
Gleason score:	
Gleason < 7	107 (31.66%)
Gleason ≥ 7	231 (68.34%)
T Stage (TNM)	
T1–T2	209 (83.60%)
T3–T4	41 (16.40%)
D’Amico Risk	
Low-Mid	157 (45.77%)
High	186 (54.23%)
Metastasis	
Yes	133 (51.55%)
No	125 (48.45%)

Some clinical variables were unavailable for certain samples, leading to a varying total n across categories.

## Data Availability

The original contributions presented in this study are included in the article/Appendix A. Further inquiries can be directed to the corresponding author(s).

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
