# Peer review of "GSTM5 as a Potential Biomarker for Treatment Resistance in Prostate Cancer"

_biomedicines, 2025, doi:10.3390/biomedicines13081872_

Round 1
Reviewer 1 Report
Comments and Suggestions for Authors
This study reported the GSTM5 as a useful biomarker for predicting treatment resistance of prostate cancer. The reviewer would like to suggest some critiques as follows.
- On line 24, “PC patients” is wrong. “patients with PC” is correct.
- On line 26, what are ”clinical risk factors” and “ADT response”?
- On line 29, the authors should state the exact number, not p<0.05.
- On line 29, what is OR
- On line 31, the authors suggest that the significant digits of the P-values should be standardized to three digits.
- On line 48, “a significant proportion” is inadequate. Please give the approximate percentage.
- On line 51, the authors should change to testosterone instead of androgen.
- D'Amico's risk classification is currently not common. The authors should change to the NCCN risk classification and redo the analysis.
- On line 88, what is “poor patients prognosis”? OS or PFS?
- On line 139, the authors should include PSA values in the grouping of patients.
Reviewer 2 Report
Comments and Suggestions for Authors
Verify results using bigger, multiethnic models.
Examine the GSH-mediated processes and the epigenetic regulation of GSTM5 in PC.
Examine treatment approaches that target the GSTM5 pathways, such as redox modulators and demethylating drugs.
Although this study offers fundamental insights into the genetic function of GSTM5 in ADT resistance, mechanistic and clinical validation are necessary to fully explore the diagnostic potential of this protein.
Comments on the Quality of English LanguageMay improve for better understanding of readers
Round 2
Reviewer 1 Report
Comments and Suggestions for Authors
none.
Author Response
Dear Reviewer 1,
We sincerely thank you for your positive feedback and for accepting the changes we made in the first revision. We truly appreciate your helpful comments, which allowed us to improve the manuscript. Since Reviewer 2 suggested some additional modifications, we have made further revisions that you are welcome to review if you wish.
Reviewer 2 Report
Comments and Suggestions for Authors
1. The authors primarily concentrate on resolving reviewer feedback and enhancing the manuscript's readability, but they omit important details like the lack of in vivo validation, possible off-target effects, and the small study population and experimental model scope which is an important aspect of any manuscript.
2. It is being observed that although the authors acknowledge the uniqueness of examining GSTM5 and its polymorphism in prostate cancer, they do not go into detail on mechanistic pathways or suggest specific follow-up studies to address open-ended problems needed for better understanding the topic.
3.The authors don't explain how their findings relate to the clinical or translational context that the journal emphasizes, nor do they offer suggestions for how to increase the article's effect or visibility (e.g., through outreach or non-technical summaries).
In conclusion, the authors' remarks partially address reviewer feedback and manuscript clarity, but they fall short in providing a thorough analysis of the ethical, mechanistic, and more general scientific or editorial factors that the journal require.
Comments on the Quality of English LanguageFine
Round 3
Reviewer 2 Report
Comments and Suggestions for Authors
Although the study is scientifically sound and fills a significant therapeutic need, the results should be regarded as preliminary due to the aforementioned limitations, particularly the small sample size, lack of external validation, and lack of functional validation.
The findings should not currently be utilized to direct clinical treatment without additional validation, but the report might be appropriate for publication as a pilot study or hypothesis-generating study.
In the discussion and conclusion, the writers should make these limits and the need for more study very evident.
Author Response
Summary
We sincerely appreciate the time and effort you have dedicated to reviewing our manuscript. Your thoughtful comments have been invaluable in helping us improve its quality. Please find below our detailed response to your observation.
Point-by-point response to Comments and Suggestions for Authors
Comment 1: Although the study is scientifically sound and fills a significant therapeutic need, the results should be regarded as preliminary due to the aforementioned limitations, particularly the small sample size, lack of external validation, and lack of functional validation.
The findings should not currently be utilized to direct clinical treatment without additional validation, but the report might be appropriate for publication as a pilot study or hypothesis-generating study.
In the discussion and conclusion, the writers should make these limits and the need for more study very evident.
Response 1:
Thank you very much for your insightful and constructive feedback. We fully agree with your comment that the findings should be regarded as preliminary, and we appreciate your suggestion to emphasize this in the manuscript.
We would like to clarify that, as you rightly pointed out, we have already addressed this important consideration in the manuscript. Specifically, in the Discussion section (lines 471-502), we have highlighted the study’s limitations, including the need for further validation, and explicitly stated that functional studies and external validation are essential for confirming the results.
“Despite the promising findings of this study, certain limitations must be considered. The small sample size limits the statistical robustness of some analyses; therefore, future studies with larger cohorts and external validation are needed to confirm the clinical relevance of GSTM5 as a predictive biomarker of ADT resistance. Additionally, although our data suggests its involvement in treatment resistance, functional studies in cellular and preclinical models will be required to elucidate the underlying molecular mechanisms [21]. Specifically, future in vivo studies should focus on investigating how GSTM5 interacts with key resistance pathways, such as oxidative stress regulation and AR signaling [2, 13-16]. Moreover, treatment resistance in PC is a multifactorial process involving diverse molecular pathways; thus, it will be crucial to investigate how GSTM5 interacts with other resistance pathways, such as alterations in BRCA2, TP53, and PTEN, which have been shown to influence therapeutic response [30]. Additionally, further functional studies are necessary to determine how GSTM5 modulates oxidative stress and its crosstalk with other signaling pathways, such as NRF2 or DNA repair mechanisms, in the context of PC and ADT resistance [31]. Although our study focuses on GSTM5, it is part of the larger GST family, which includes GSTM1 and GSTM2, both of which have been implicated in tumor progression and treatment response [19, 20]. This underscores the importance of considering other members of the GST family in future studies that adopt more integrative approaches. In clinical practice, genotyping this variant before initiating ADT could help physicians identify patients more likely to benefit from standard hormonal therapy versus those who may require early intensification strategies, including the addition of second-line agents. These preliminary findings lay the groundwork for future clinical trials that incorporate GSTM5 status as a stratification variable or therapeutic target.
From a translational perspective, these preliminary findings open new avenues for personalizing the management of advanced PC. In clinical practice, genotyping the rs3768490 polymorphism prior to initiating ADT could assist physicians in identifying patients more likely to benefit from standard hormonal treatment versus those who may require early intensification strategies, such as the addition of second-line agents. Prospective studies are warranted to evaluate whether GSTM5 genotyping can be effectively integrated into clinical workflows, enabling earlier identification of patients at higher risk of developing resistance to ADT.”
Additionally, in the Conclusion (lines 516-518), we briefly reiterate the need for future research and validation.
“Further functional and clinical validation is warranted to confirm its utility and explore potential therapeutic interventions targeting redox balance and epigenetic regulation in PC”.
We believe that these sections clearly convey our position on the preliminary nature of the results and the necessity for additional studies, as well as the specific types of studies required.
Round 4
Reviewer 2 Report
Comments and Suggestions for Authors
- It is observed in the manuscript that while TCGA data suggested GSTM5 promoter hypermethylation in prostate cancer, the study did not experimentally validate this in the patient cohort or link it to the rs3768490 genotype which is important.
- The study proposes rs3768490 as a predictive biomarker but offers no validated risk-stratification model, clinical utility assessment, or integration pathway into existing decision-making frameworks like the D'Amico classification which should be added to the manuscript.
Must be improved
